# Topological Distribution of Wound Stiffness Modulates Wound-Induced Hair Follicle Neogenesis

**DOI:** 10.3390/pharmaceutics14091926

**Published:** 2022-09-12

**Authors:** Hans I-Chen Harn, Po-Yuan Chiu, Chein-Hong Lin, Hung-Yang Chen, Yung-Chih Lai, Fu-Shiuan Yang, Chia-Ching Wu, Ming-Jer Tang, Cheng-Ming Chuong, Michael W. Hughes

**Affiliations:** 1International Research Center for Wound Repair & Regeneration, College of Medicine, National Cheng Kung University, Tainan 70101, Taiwan; 2Department of Pathology, Keck School of Medicine, University of Southern California, Los Angeles, CA 90007, USA; 3Institute of Clinical Medicine, College of Medicine, National Cheng Kung University, Tainan 70101, Taiwan; 4Institute of Basic Medicine, College of Medicine, National Cheng Kung University, Tainan 70101, Taiwan; 5Department of Mechanical Engineering, National Chung Hsing University, Taichung 40227, Taiwan; 6Department of Medical Research, China Medical University Hospital, Taichung 404327, Taiwan; 7Institute of Translational Medicine and New Drug Development, China Medical University, Taichung 40402, Taiwan; 8Department of Cell Biology and Anatomy, College of Medicine, National Cheng Kung University, Tainan 70101, Taiwan; 9Department of Physiology, College of Medicine, National Cheng Kung University, Tainan 70101, Taiwan; 10Department of Life Sciences, National Cheng Kung University, Tainan 70101, Taiwan

**Keywords:** wound healing, full-thickness, regeneration, hair follicle, wound-induced hair follicle neogenesis, mechanotransduction, stiffness, atomic force microscopy, microenvironment

## Abstract

In the large full-thickness mouse skin regeneration model, wound-induced hair neogenesis (WIHN) occurs in the wound center. This implies a spatial regulation of hair regeneration. The role of mechanotransduction during tissue regeneration is poorly understood. Here, we created wounds with equal area but different shapes to understand if perturbing mechanical forces change the area and quantity of de novo hair regeneration. Atomic force microscopy of wound stiffness demonstrated a stiffness gradient across the wound with the wound center softer than the margin. Reducing mechanotransduction signals using FAK or myosin II inhibitors significantly increased WIHN and, conversely, enhancing these signals with an actin stabilizer reduced WIHN. Here, α-SMA was downregulated in FAK inhibitor-treated wounds and lowered wound stiffness. Wound center epithelial cells exhibited a spherical morphology relative to wound margin cells. Differential gene expression analysis of FAK inhibitor-treated wound RNAseq data showed that cytoskeleton-, integrin-, and matrix-associated genes were downregulated, while hair follicular neogenesis, cell proliferation, and cell signaling genes were upregulated. Immunohistochemistry staining showed that FAK inhibition increased pSTAT3 nuclear staining in the regenerative wound center, implying enhanced signaling for hair follicular neogenesis. These findings suggest that controlling wound stiffness modulates tissue regeneration encompassing epithelial competence, tissue patterning, and regeneration during wound healing.

## 1. Introduction

Tissue regeneration of a hair follicle with complex architecture is not trivial, and requires multiple tissues including epithelium, dermis, muscle, and nerve. In 1954, Dr. Breedis described de novo hair regeneration after large full-thickness (LFT) wounding [1], and decades later wound-induced hair follicle neogenesis (WIHN) was characterized for the mouse [2]. The WIHN is a powerful assay for studying de novo skin organ regeneration after full-thickness trauma. Multiple studies have investigated the mechanism of WIHN, namely the requirement for Wnt/β-catenin [2], derivation of FGF9 from γδT-cells [3], the activation of TLR3 leading to p-STAT3 induction of Wnt/β-catenin [4], epithelial competency [5], and tissue mechanics [6]. Interestingly, mouse hairs regenerate only in the wound center [2,3,4], and humans fail to regenerate resulting in scar [7]. Scar tissue does not contain ectodermal organs or rete ridges and is relatively stiff compared to normal tissue [8]; this stiffness could play a role in regulating tissue regeneration.

Human terminal hair is similar to mouse primary follicles, and human vellus hair is similar to mouse secondary follicles. Humans do not regenerate terminal or vellus hair after LFT wound healing, instead forming scar tissue, and this is a significant clinical issue [9]. Scar tissue is more stiff than normal tissue and can be reduced by modulating focal adhesion kinase (FAK) signaling [10]. The laboratory mouse C57BL/6 regenerates hair follicles after LFT wounding in the center of the wound bed but not at the margin during WIHN [2,3,4,5,6]. Additionally, these mice only regenerate secondary hair follicles, specifically zigzag [2,11]. Secondary hair follicles are less complex than primary follicles [12]. This raises many questions, for example whether there are inhibitors expressed from the wound edge that inhibit WIHN or if is there another mechanism at work. Is there a minimum distance between the wound center and margin for WIHN and tissue regeneration? Why does WIHN only generate zigzag follicles? Interestingly, African spiny mice can regenerate primary and secondary hair follicles, including all four hair shaft types [11]. Furthermore, they can regenerate these hair follicles in the wound margin, unlike C57BL/6 mice. This extraordinary regeneration ability is not restricted to the skin organ. Ear cartilage, skeletal muscle, spinal cord, and kidneys also exhibit higher regenerative ability [13]. A clue to how this ability works is that African spiny mice exhibit softer skin tissue and this associates with greater regenerative ability [6,14]. Therefore, we surmised a gradient could be established in the wound bed that modulates tissue regeneration and patterning. Utilizing C57BL/6 mice, we studied how initial wound shape affected tissue regeneration, WIHN, and hair patterning by increasing the distance between the wound margin and center. Our hypothesis is that, when increasing the distance from the wound margin to the wound center during LFT wound healing, both WIHN and tissue regeneration would be increased.

Here we confirm a stiffness gradient in the LFT wound bed, low in the center and high in the margin, and that manipulating wound stiffness physically or chemically regulates regenerative potential. This result partly answers the question of why hair regenerates in the wound center. Previous reports described a role for FAK during scar formation [8,10,15], and here we show a role for the FAK/SMA/MyoII axis during skin regeneration after LFT injury. Finally, these results demonstrate mechanobiological cues are sensed throughout the wound bed to regulate tissue regeneration through p-STAT3 induction. In summary, we were able to control hair follicle neogenesis in LFT wounds by specifically modulating the wound bed stiffness. We studied the wound healing responses of ‘repair’, namely re-epithelialization with scar formation resulting in loss of structure and function, and ‘regeneration’, the restoration of ectodermal organs, tissue integrity, and function to near normal. The need to understand cellular and molecular mechanisms modulating wound healing and tissue regeneration is imperative. Regenerative medicine therapies can be developed targeting the ability to activate regenerative signals that simultaneously inhibit scar formation and enhance regeneration after LFT severe trauma.

## 2. Materials and Methods

### 2.1. Animal Models

All animal work was performed according to the approved animal protocol number 109292, approved 12 August 2020, in accordance with the guidelines and regulations for the care and use of laboratory animals at National Cheng Kung University. All mice were housed in climate-controlled indoor facilities. The C56BL/6J mice were purchased from the National Cheng Kung University Animal Center. For live in vivo imaging, Gt(ROSA)26Sortm4(ACTB-tdTomato-EGFP)^Luo/J^ mice purchased from Jackson Laboratory were utilized.

### 2.2. Wound-Induced Hair Neogenesis Assay

The WHIN assay was previously described [2]. Briefly, a 1 cm^2^ square, 1 cm^2^ diamond, or 1 cm^2^ circle full-thickness wound was excised on the posterior dorsum of 3-week-old mice and observed for hair neogenesis. Mice were anesthetized using isoflurane, full-thickness skin was excised, and the analgesic ketorolac (12 μg/g b.w.) was given by intraperitoneal injection (IP). To create the diamond wound, a 1 cm^2^ square wound was excised at a 45-degree rotation relative to the square wound. To create the circle wound, a circle with radius 0.564 cm was excised. All wound shapes contained equal excised areas of 1 cm^2^.

### 2.3. Inhibitor Treatments

The PF573228, blebbistatin, or jasplakinolide (Cayman, Ann Arbor, MI, USA) were dissolved in DMSO. Then, 20 μL was applied once per day directly to the wound surface, starting at PWD 10 and continuing until PWD 17.

### 2.4. Separation of Epidermis and Dermis

The epidermis was separated from the dermis of wounds as previously reported [2]. Briefly, full-thickness wounds were incubated in 20 mm EDTA in phosphate buffered saline (PBS) at 37 °C overnight. The next day, the epithelium was gently peeled away from the dermis with fine watchmaker’s forceps.

### 2.5. Alkaline Phosphatase (ALP) Stain

To detect newly formed dermal papillae, alkaline phosphatase (ALP) staining was performed as previously reported [2]. Briefly, full-thickness wounds were excised, the and epidermis was separated from the dermis using 20 mM EDTA. The dermis was fixed in acetone overnight at 4 °C and washed in PBS several times. The dermis was preincubated in ALP buffer (0.1 M Tris-HCl, 0.1 M NaCl, 5 mM MgCl_2_, and 0.1% Tween 20) for 30 min, and then incubated with BCIP/NBT color development substrate (Promega) in ALP buffer at 37 °C until color development. The reaction was stopped by washing with pH 8.0 Tris-EDTA, and the tissue was stored in PBS with sodium azide.

### 2.6. Atomic Force Microscopy (AFM)

In this study, AFM (Nanowizard II, JPK, Berlin, Germany) was set up for contact mode indentation in PBS. The spring constants of all cantilevers were calibrated via the thermal noise method with a correction factor in liquid [16,17] prior to each measurement, and were valued at 0.08 N/m. To overcome the unevenness of skin wound tissue, we attached a large polystyrene bead (25 μm in diameter) on tipless cantilevers (CSC12-F, MikroMasch, Wetzlar, Germany) and applied a high indenting force at 5 nN. This value showed the least discrepancy after pre-testing using 1, 3, 5, and 10 nN with the same cantilever on test samples. The approaching and retracting rates of cantilever were set at 1 μm/s. Force–distance curves were collected and calculated with the JPK package software, which was based on the Hertz model with a spherical indentation [18,19]. The force on the cantilever F(h) is calculated as follows:$$F\left(h\right)={\frac{E_{\mathrm{sample}}}{{1-v}_{\mathrm{sample}}^{2}}\frac{4\sqrt{R}}{3}h}^{3/2}$$
where h is the depth of the indentation, E is the effective modulus of a system tip-sample, v is the Poisson ratio for the sample, and R is the radius of the AFM tip. The unit of Young’s modulus is calculated as N/m^2^, and expressed as pascal (Pa) or kilopascal (kPa). The Poisson ratio was set at 0.5, since the spherical tip was incompressible relative to the sample. The temperature of the measurement was controlled at 32 °C to mimic the surface temperate of mouse skin [20]. To maintain the biomechanical force integrity of the dorsal wound, the entire mouse skin organ was removed by creating an excision on the ventral side midline, extending from the anterior neck region to the posterior groin region, and then dissecting away the skin organ from the underlying fascia. Normal skin and wound stiffness were immediately measured after skin organ removal to prevent artifacts from tissue decomposition. In order to measure the stiffness of the entire wound, AFM measurements were positioned across the wound starting from unwounded normal skin on one side and progressively traveling through to the opposite side, via the near wound edge, the near wound bed, the wound center, the opposite wound bed, the opposite wound edge, and the opposite normal skin. At least five indentation points were taken for each region of interest.

### 2.7. Heatmap of Spatial Stiffness of the Wound

The interpolation of Young’s modulus was performed by using the 3D meshgrid function of MATLAB. After obtaining a Young’s modulus (z) at a specific spatial location (x, y) in the wound, a three-dimensional matrix was defined. When the positions and Young’s modulus of all the measured spots were identified, we could interpolate the Young’s modulus of the positions in between to average the Young’s modulus of the nearest parameters using the 3D meshgrid function, by defining (x, y) as the meshgrid and (z) as the griddata. In the end, the heatmap was generated by defining the representative color of Young’s modulus.

### 2.8. Live in vivo Imaging of the Wound

A 1 cm^2^ square wound was excised from a 3-week-old (p21) ROSA^mTmG^ mouse. The mouse was anesthetized using inhaled isoflurane and the wound was observed at PWD 17 with an inverted confocal microscope (FV-1000, Olympus) utilizing a 40× or 63× oil lens. The viewing medium between the wound and the cover slide was 50% glycerol in PBS. Then, DMSO with or without FAK-inhibitor (FAK-I) was added for real-time monitoring of cell shape. A Z-stack series of 20 slices was recorded during a time course starting at the cornified layer of the epidermis and ending at a 50 μm depth for each time point.

### 2.9. Image Quantification and Analysis

The quantification of image area, wound geometries, cell aspect ratio, and orientation were analyzed using ImageJ according to the user guide ImageJ/Fiji 1.46.

### 2.10. Histology and Immunostaining

The wound tissues were fixed in 4% PFA and dehydrated in a graded alcohol series. The tissue was cleared in xylene and embedded in paraffin wax. Six-micron sections were cut on a microtome. Then, H&E sections were performed according to accepted protocol. Whole-mount tissues were fixed in 4% PFA and then stored at 4 °C in PBS with NaAzide. Paraffin section and whole-mount immunostaining were performed as previously described [21]. Briefly, fixed tissues were permeabilized with methanol and blocked with 3% H_2_O_2_ for 30 min, and then serum blocked for 1 h. The primary antibody was added and incubated over night at 4 °C with agitation. The tissue was washed with TBST (Tris-buffered saline Tween 20) and the secondary antibody was added for 1 h at room temperature. The tissue was washed with TBST and, if utilized, a tertiary antibody was added for 1 h at room temperature. The tissue was washed, and color was developed using the AEC kit (Vector Laboratories, Burlingame, CA, USA), or fluorescence was visualized with a fluorescence microscope. The whole-mount samples were cleared in a series of glycerol-PBS until 100% glycerol for imaging. The α-SMA antibody is from Thermofisher (Thermofisher, Waltham, MA, USA), the FAK, the pY397-FAK, the EPHA3, and the p-Tyr705-STAT3 antibodies and alexa-488 secondary antibody are from Abcam (Abcam, Waltham, MA, USA). The pSTAT3 sections were counterstained with eosin.

### 2.11. Tissue Culture, Western Blot, and Immunoprecipitation Blot

To perform tissue culture of the epidermis, the wound was harvested on PWD 14, and the epidermis was isolated under a dissecting microscope and placed in a culture dish with minimal keratinocyte-SFM (Thermofisher, Waltham, MA, USA) plus 10% FBS added for 2 h to allow attachment. FAK-I or DMSO was added to 1 mL of the medium another 6 h. To collect tissue lysate, 250 μL of lysis buffer was added to the dish, and tissue slurry was collected and grinded using a homogenizer in a 1.5 mL microtube. To harvest the tissue directly, the wound was harvested at PWD 14, microdissected, and immediately placed in liquid nitrogen for 30 s. The samples were then grinded using a homogenizer in a 1.5 mL microtube with 500 μL of lysis buffer. The Bio-Rad General Protocol for Western Blotting (Bio-Rad) was followed. The membrane was hybridized with α-SMA (Thermofisher, Waltham, MA, USA), FAK (Abcam, Waltham, MA, USA), pY397-FAK (Abcam, Waltham, MA, USA), GAPDH (Abcam, Waltham, MA, USA) β-actin (Abcam, Waltham, MA, USA), and EPHA3 (Abcam, Waltham, MA, USA) primary antibody at 4 °C overnight, and then with the secondary for 1 h at room temperature. Immunoprecipitation (IP) was performed according to accepted protocol. Briefly, an antibody for the protein of interest was incubated with A/G-couple agarose beads (Millipore, Billerica, MA, USA) at room temperature for 30 min. Target protein was incubated with the antibody–bead complex at 4 °C overnight. The target protein complex was isolated and separated using the aforementioned western blot assay.

### 2.12. RNA Extraction and RNA-seq

The RNA-seq was performed on replicate samples from treated or control 1 cm^2^ square PWD 14 wounds, and the samples were further dissected into wound margin and wound center. The wound was harvested, and a 3 mm diameter hole-punch biopsy was taken from the geometric center of the wound to identify the wound center, and the remaining wound tissue was considered the margin. The epidermis and dermis were separated manually under a dissecting microscope. The dissected epidermis tissues were placed in liquid nitrogen, disaggregated individually using a mortar and pestle, and then collected into a 1.5 mL microtube. The RNA was extracted using the RNeasy Mini Kit (QIAGEN, Hilden, Germany). Then, 1 μg of total RNA from each sample was used to construct an RNA-seq library using the TruSeq RNA sample preparation v2 kit (Illumina, San Diego, CA, USA). Sequencing (50 cycles double read) was performed by Welgene (Taipei, Taiwan) using Nextseq (Illumina Solexa, San Diego, CA, USA).

### 2.13. RNA-seq Analysis

The mouse mm9 reference genome, and RefSeq genome annotation downloaded from the UCSC Genome Browser were used for RNA-Seq analysis [22]. The alignment, quantification, normalization, and differential expression analysis were performed by TopHat2 [23], Partek E/M (Partek Inc, Saint Louis, MO, USA), TMM [24], and GSA [25], respectively. The false discovery rate < 0.05 was set as a threshold to identify differentially expressed genes. The principal component analysis (PCA), hierarchical clustering, and Venn diagrams were carried out using the Partek Genomics Suite 6.15.1207 (Partek Inc. Saint Louis, MO, USA). The RNAseq data has been uploaded to the Gene Expression Omnibus website. The sample accession numbers are as follows: GSM4850917, GSM4850918, GSM4850919, GSM4850920.

### 2.14. qPCR

Reverse-transcription was performed using SuperScript II Reverse Transcriptase (Invitrogen, Thermo Fisher Scientific, Waltham, MA, USA). In brief, 200 ng of mRNA was added into 500 μg of Oligo(dT), and 10 mM of dNTP Mix, diluted in distilled water to a final volume of 12 μL. The mixture was heated to 65 °C for 5 sec and then quickly chilled on ice. Then, 4 μL of 5X first-strand buffer, 2 μL of 0.1 M DTT, and 1 μL of RNaseOUT (40 units/μL) were added into the tube. After 42 °C incubation for 2 min, 1 μL (200 units) of SuperScript II RT was added, and the mixture was incubated again at 42 °C for 50 min. The reaction was inactivated by heating at 70 °C for 15 min. Then, SYBR Green Master Mix was used to perform qPCR. In brief, SYBR Green Master Mix was mixed with 1 μM of forward and reverse primers and 200 ng of cDNA. All reactions were performed in 20 μL total volume with triplicates (*n* = 3). The samples were incubated (Mx3000P, Stratagene, San Diego, CA, USA) at 50 °C for 2 min, 95 °C for 2 min, and then cycled 40 times between 95 °C for 15 secs and 60 °C for 1 min. The primers were designed in reference to PrimerBank—MGH-PGA (Boston, MA, USA). The amplification efficiency was calculated using the slope of the regression line of the standard curve. Expressed genes were normalized to the reference gene GAPDH within the same sample to determine dCt. This dCt was then exponentially transformed into dCt expression. After averaging the triplicates, the value was then normalized to the averaged control value ddCt.

### 2.15. Statistics

The photographs are representative samples of at least four replicates. Hair follicle counts are reported as the average from at least four samples. Each bar on the qPCR graph represents the average and SEM of three independent samples. All data is presented as mean ± SEM unless stated otherwise. Results from a Student’s *t*-test with *p* < 0.05 were considered significant. Here,* *p* < 0.05, ** *p* < 0.01 and *** *p* < 0.001.

## 3. Results

### 3.1. Wound Shape Corresponded with Hair Regeneration Patterns

The pattern of hair regeneration occurring only the in the wound center, with a wound bed margin lacking regeneration, suggested a secreted inhibitor gradient, high in the margin and low in the center. We hypothesized that an increase in distance from the wound center to the margin would increase WIHN. To investigate this hypothesis during WIHN, 1 cm^2^ wounds were created in square, diamond, and circle shapes on the dorsal skin of 3-week-old (p21) C57BL/6J mice (Figure 1a). The healed wounds were observed every seven days (Appendix A) and the location of newly formed hair follicle patterns with respect to wound shape on post wound day (PWD) 28 are illustrated (Figure 1b,c and Appendix A–g). Here, WIHN occurred in the wound center surrounded by a non-hair forming region at the wound margin in all shapes (Figure 1c,d). The square wound healed with a WIHN pattern of a small rectangle, the diamond healed with a WIHN pattern of a thin rhombus, and the circle wound healed a WIHN pattern of a short column (Figure 1b,c). Wound contraction in all shapes was greater in the medial/lateral (~58%) than the anterior/posterior (~35%) direction (Figure 1b–d, Table 1). The circle wound exhibited the greatest change in wound parameters, namely area (−73.1%), wound length (−44.9%), and width (−61.3%), resulting in the smallest change in wound aspect ratio (142%) (Table 1, Appendix A). Interestingly, the circle wound exhibited greater medial/lateral contraction at the wound anterior end versus posterior, and resulted in the shifting of neogenic follicles posteriorly (Figure 1b,c). The number of regenerated hair follicles is significantly lower in the circle wound (10.4 ± 3.1) compared to the square (21.8 ± 2.8) or diamond (22.4 ± 4.2), and the healed wound area is significantly less in the circle wound (26.9 ± 3 mm^2^) compared to the square (36.1 ± 2.2 mm^2^) or diamond (38.3 ± 3 mm^2^) (Figure 1d). Of note, although diamond wounds healed with a longer length and width (Table 1) resulting in an increase in distance of the wound center to margin, the number and spatial distribution of neogenic follicles were not significantly different from square wounds (Figure 1c,d). This data refuted our initial hypothesis. A minimal distance of 1.4 mm was measured between the edge of the WIHN regeneration zone and the wound edge (Appendix A). In summary, all wounds contained the same initial area, but the circle shape yielded a significantly different regenerative outcome, implying that the physical wound environment contributes to regenerative ability and the spatial distribution of WIHN.

### 3.2. Regeneration Associated with Low Wound Tissue Stiffness

According to the wound shape data, wound shape corresponded to WIHN level and suggested that a mechanical microenvironment may play a role. To determine the spatial distribution of mechanical force in the skin, atomic force microscopy (AFM) was utilized to measure stiffness (Young’s modulus) during the resting phase (telogen) and growing phase (anagen) of the hair cycle in unwounded dorsal skin. Interestingly, the stiffness of anagen phase skin (~7 kPa) was significantly lower than telogen phase skin (~27 kPa) (Appendix A). The lower stiffness during anagen is not surprising because hair follicles growing into the dermis require lower adhesion, permitting stem cell motility [6,26]. Next, we measured the stiffness of the wound and adjacent skin across the square, diamond, and circle wounds (Figure 2a–l). Of note, the stiffness of the regenerative centers of square (~10.7 kPa), diamond (11.3 kPa), and circle (~13.6 kPa) wounds were more similar to anagen (~7 kPa) versus telogen (~27 kPa), and the circle wound center exhibited the highest stiffness with the lowest WIHN level (Figure 2d,h,l and Appendix A). All wound margins possessed higher stiffness (~25–43 kPa) relative to their respective wound centers (~10–13 kPa) (Figure 2b,f,j). Heat maps generated via spatial wound stiffness measurements for each PWD 14 wound shape demonstrated lower stiffness in the wound center (blue, Figure 2c,g,k) and higher stiffness in the wound margin (red, Figure 2c,g,k). Interestingly, the site with the lowest stiffness for the circle wound coincides with the widest part of the circle wound and associates with the neogenic follicles (Figure 1b,c,o). These correlations suggest that WIHN favors a low stiffness environment, located in the center and distant from the wound margin, and is in agreement with previous studies [2,3,4,6]. Consequently, the hypothesis was revised to state that homeostatic regeneration, wound tissue regeneration, and WIHN requires a relatively soft microenvironment, and this is inhibited by increasing stiffness.

### 3.3. Wound Keratinocyte Cell Aspect Ratio Reflects Wound Stiffness

Prior studies have shown cell shape reflects the local microenvironment [27,28,29,30]. Epidermal cell aspect ratios from different PWD 14 wound bed regions were compared (Appendix A). The lowest average cell aspect ratio was found in the wound center (<1.5) and the highest was found in the wound margin (>1.75) (Appendix A). Whole mount immunostaining of wound epidermis showed lower α-SMA expression in the wound center and higher in the wound margin. Wound margin α-SMA positive cells orientated the cellular longitudinal axis towards the wound center (Appendix A). These results correspond with the AFM measurements and show wound stiffness is spatially regulated, suggesting an important role in modulating regenerative ability and patterning.

### 3.4. FAK Phosphorylation Correlates with SMA Influencing Topological Distribution of Wound Stiffness

In order to understand how FAK-I treatment affected wound stiffness, AFM, western blotting, and immunohistochemistry were performed. Phosphorylation of focal adhesion kinase (FAK) at tyrosine-Y397 is a key mediator of mechanotransduction during mechanical stimulation [31]. Re-epithelized wounds were treated with 1 mM FAK-I at PWD 10, and the wound tissue was harvested at PWD 14 (Figure 3a). Then, AFM was used to measure wound stiffness after topical FAK inhibition treatment (Figure 3b,c). The FAK inhibition led to an overall reduction in wound stiffness with a concomitant increase in wound center area exhibiting lower stiffness (< 10 kPa) (Figure 3b,c blue area). Of note, this reduction in wound center area stiffness corresponded with increased WIHN (Figure 3b,c, white dots) and an increased total regenerative area upon FAK inhibition (Figure 3b,c). Live in vivo confocal imaging (4D imaging) demonstrated a decrease in the aspect ratio of epithelial cells from 1.64 ± 0.05 to 1.48 ± 0.05 during 1 M FAK-I treatment supporting FAK tissue stiffness modulation (Appendix A). The α-SMA correlated with regenerating fat cells [32] and scar contraction [10] and, thus, we compared the expression of α-SMA in wound epidermis with or without PWD 14 FAK-I treatment. Western blotting showed that FAK-I treatment downregulated FAK-pY397 in ex vivo wound epidermis and α-SMA in in vivo wounds (Figure 3d). Immunostaining of α-SMA exhibited significant expression throughout the vehicle treated wounds (Figure 3e,f). Interestingly, a low dose (1 μM) topical FAK–I treatment downregulated α-SMA expression in the wound center epithelium, with low amounts expressed near wound margins and dermal vasculature (Figure 3g,h). Additionally, p-Tyr705-STAT3, an inducer of the Wnt pathway and WIHN [4], was increased in topical FAK-I treated wound centers compared to controls (Figure 3i,j). Taken together, these results suggest FAK inhibitor treatment reduced wound stiffness, lowered mechanotransduction, downregulated α-SMA, and increased pSTAT3 in the wound, promoting a regenerative microenvironment.

### 3.5. Bioinformatic Analyses Identify Gene Networks Modulated by Mechanobiological Signaling in the Wound Microenvironment

To understand how FAK inhibition altered the wound microenvironment and increased tissue regeneration, RNA-seq was utilized to identify gene expression patterns in square wound epidermis at PWD 14. The epidermis from control and FAK inhibitor treated wounds were dissected into non-regenerative wound beds or regenerative wound centers using a 3 mm hole punch biopsy (Figure 4). The 3 mm size was chosen because the regenerative area can be accurately dissected from the non-regenerative wound area (Table 1). The following wound samples were taken: control wound bed (stiffest and non-regenerative), FAK-I treated wound bed (stiff and non-regenerative), control wound center (soft and normal regeneration), and the FAK-I treated wound center (softest and increased regeneration) were sent for RNA-seq (Figure 4a). Principal component analysis (PCA) identified 4 unique groups according to wound tissue and treatment type (Appendix A). Differential gene expression between similar wound regions with or without FAK-I treatment was used to identify candidate genes contributing to the wound microenvironment. A total of 856 genes were identified between control versus FAK-I treated wound centers, and 938 genes were identified between control versus FAK-I treated wound margins (Figure 4b). A Venn diagram identified 195 common genes between the wound center and margin that change during WIHN upon FAK-I treatment (Figure 4b). Indeed, 80% of these genes showed a similar trend of gene expression (Appendix A), suggesting that these genes were modulated by FAK-I treatment. Gene ontology analysis identified modulated cell functions, namely cell proliferation, extracellular matrix (ECM), integrin signaling, FAK signaling and Wnt/-catenin signaling (Figure 4c). Specifically, downregulation of integrins and ECM related genes with the simultaneous upregulation of cell proliferation and hair folliculogenesis genes, (*Cdk*, *Cdc*, *Lgr6*, *Wnt16*, and *Egfr*) were modulated by FAK-I treatment (Figure 4d). The qPCR validated a subset of RNAseq data (Appendix A). In order to identify specific WIHN enhancing genes targeted by FAK-I treatment, control non-regenerative wound margins versus treated regenerative wound centers were analyzed (Figure 4e,f). Differential gene expression analysis of FAK-I treated center versus control margin wound epithelia identified *Epha3* upregulation (Figure 4e). This is interesting because ephrin receptors are protein–tyrosine kinases located at the cell surface, which coordinate cell–cell interactions and are known to phosphorylate STAT3 [33]; it is known that pSTAT3 induces WIHN [4]. This led us to ask if EPHA3 physically associated with pSTAT3 in the wound center. Western blots of microdissected wounds showed increased EPHA3 in the wound center versus the margin, supporting the RNAseq data (Appendix A). Additionally, immunoblots of microdissected wounds demonstrated that EPHA3 immunoprecipitated pSTAT3 in the wound center, suggesting direct phosphorylation (Appendix A). Gene set enrichment analysis of the FAK-I treated wound center versus the control margin wound epithelium showed upregulation of tissue morphology, cell movement, and cell survival with the simultaneous downregulation of cell death functions (Figure 4f). This RNAseq data suggests FAK-I treatment affects multiple gene networks related to the biomechanical environment and helps to elucidate how WIHN is promoted.

### 3.6. Mechanobiological Signaling Modulates Wound-Induced Hair Follicle Neogenesis

The RNAseq data suggested that FAK was altering the wound mechanobiological microenvironment. Therefore, we predicted that other constituents of the mechanical microenvironment could be targeted to regulate regeneration and WIHN. To understand this role of mechanotransduction during WIHN, different inhibitors were topically applied to the healing wounds in a dose–response assay. First, FAK inhibitor PF573228 (FAK-I) was applied topically to the surface of 1 cm^2^ square wounds once per day in a dose response from PWD 10 to PWD 17, and WIHN was analyzed on PWD 28 (Figure 5a). PWD 10 was the approximate day the fibrin clot passively desquamates from the wound after complete reepithelization. The vehicle (DMSO)-treated wounds regenerated 16.8 ± 3.8 hair follicles, with low dose FAK-I (1 μM) 23.1 ± 4.1, and high dose FAK-I (10 μM) 4.7 ± 1.3 (Figure 5b–e). Next, the actin cytoskeleton, a downstream component of FAK during mechanotransduction, was targeted by treating square wounds with blebbistatin, an inhibitor of myosin II activity in a dose-response (Figure 5a). A moderate-dose blebbistatin treatment (100 μM) significantly increased WIHN (57.9 ± 5.2) compared to vehicle (31.7 ± 3.1) (Figure 5f–i). Since both FAK-I and blebbistatin treatments reduce intracellular force generation [34,35], we hypothesized that an increase in intracellular force generation would reduce WIHN. Square wounds were treated with jasplakinolide, an actin polymerization stabilizer, in a dose–response experiment to enhance cellular forces (Figure 5a). Low-dose jasplakinolide (50 μM) significantly decreased WIHN (13.7 ± 5.8) compared to vehicle (34.7 ± 2.5) (Figure 5j–m). Thus, reducing cellular forces enhanced WIHN and increasing cellular forces inhibited WIHN (Figure 5n). Additionally, regenerative wound area (green line, Figure 5b–d,f–h,j–l, area occupied by neogenic follicles) significantly changed after treatment. Both FAK-I and blebbistatin treatment significantly expanded regenerative wound areas, while Jasplakinolide significantly reduced them (Figure 5o and Appendix A–h). Interestingly, all three inhibitors, FAK-I, blebbistatin, and jasplakinolide, altered the final regenerative area shapes relative to controls (Figure 5b–d,f–h,j–l versus Figure 1). These data suggest that by modulating FAK/SMA/MyoII, key players of mechanotransduction, the level of WIHN, regenerative area size, and WIHN patterning were controlled (Appendix A).

## 4. Discussion

The original hypothesis that increasing the distance between the wound edge and the regenerative wound center would increase WIHN was refuted by the data. Our revised hypothesis is that homeostatic regeneration, wound tissue regeneration, and WIHN requires a relatively soft microenvironment, and this is inhibited by increasing stiffness. In this study, the data show that wound stiffness modulates WIHN and is partially regulated through mechanotransduction pathways. The WIHN presents in the wound center as being associated with the soft microenvironment (Figure 1 and Figure 2), lower FAK activity (Figure 3), lower α-SMA expression (Figure 3), higher pSTAT3 (Figure 3), reduced cell aspect ratio (Appendix A), lower ECM expression (Figure 4), lower cytoskeletal signaling (Figure 4), increased cell survival (Figure 4), and higher EPHA3 (Appendix A). Furthermore, EPHA3 associates with pSTAT3 (Appendix A), which leads to increased WHIN signaling (Figure 4), and targeting of the FAK/SMA/MyoII axis modulates WIHN (Figure 5). These data support our hypothesis for relatively softer wound microenvironments promoting tissue regeneration both homeostatic and after injury. Furthermore, multiple targets for developing novel regenerative medicine clinical therapies are suggested by the data, including FAK, SMA, MyoII, pSTAT3, and EPHA3.

Regeneration occurs only in the wound center and not at the margin [2,3,4,5,6]. The current study shows that WIHN only presented in the wound center, and that it corresponds to the following: softer wound stiffness, lower wound FAK activity, lower epidermal α-SMA expression, higher epidermal pSTAT3 expression, lower epithelial cell aspect ratio, lower ECM expression, lower cytoskeletal signaling, and higher hair neogenesis signaling (Figure 1, Figure 2, Figure 3, Figure 4 and Figure 5). These findings concur with previous WIHN studies [2,3,4,5,6]. Conversely, higher stiffness at the wound center resulted in less WIHN (Figure 2, Figure 3 and Figure 4). The study by Nelson et al. (2015) shows that WIHN is enhanced via activation of the TLR3 receptor due to increased dsRNA [4]. This was achieved through a modified WIHN assay where micro-cuts were perpendicularly created along the original 1 cm^2^ wound excision border. These micro-cuts can disrupt contraction, resulting in a softer wound bed enhancing WIHN. The current study’s finding of increased regeneration area and WIHN due to softer wounds is similar and delineates a potential mechanism through the cytoskeletal network and adhesion complex signaling. The RNA-seq analysis data of differential gene expression and the EPHA3 IP data between center and margin of PWD 14 wound epidermis support this observation (Figure 4 and Appendix A). The FAK inhibitor treatment resulted in downregulation of ECM, integrin, and cell death gene expression, and the upregulation of cell motility, survival, and ephrin gene expression, which fosters a softer environment to enhance cell invagination and promotes WIHN. In addition, cell aspect ratio analysis shows cells in the wound edges are significantly higher than the wound center (Appendix A) suggesting that cells residing in the wound margin experience a different mechanical stimulation from the wound center, modulating the production of hair neogenesis-related morphogens, for example Wnt or Shh. Taken together, this study proposes that mechanotransduction partly explains why WIHN occurs only in the wound center.

Seifert et al. and Hans et al. show that spiny (Acomys) mice regenerate hair with greater ability than C57Bl/6 mice due in part to softer skin environments [6,14]. Stress–strain curves show that spiny mice possess lower tensile skin strength than C57Bl/6 mice [14]. Atomic force microscopy measurements of healed wounds show that spiny mice hair regenerating wound margins are significantly less stiff than C57Bl/6 mice hairless margins [6]. This manuscript’s data is in accordance with both Seifert et al. and Hans et al., suggesting that a soft wound microenvironment promotes tissue regeneration. Interestingly, spiny mice can regenerate primary and secondary hair follicles, including all four hair follicle types of guard, awl, auchene, and zigzag [11]. Even though C57BL6 mouse wound center stiffness is very similar to spiny mice [6] and this regenerative area can be increased for WIHN (Figure 3 and Figure 5), these mice cannot regenerate all hair types and only regenerate secondary follicles [2,5,11]. This is a significant clinical issue, because human terminal hair is more similar to mouse primary follicles, and regional specific hair follicle regeneration requires further study.

Wong et al. demonstrate wound stretching increases tension and scarring, a pro-inflammatory process [8]. The high skin tension induces FAK activity by recruiting immune cells, thus, mediating scarring. The current study shows that WIHN favors a physical environment resembling the relatively soft skin stiffness of anagen (Figure 2 and Appendix A). Furthermore, our data show increasing stiffness via jasplakinolide significantly reduces regenerative total area and WIHN (Figure 5). The relative stiffness of hair cycling skin changes from ~27 kPa during telogen to ~7 kPa during anagen (Appendix A). Interestingly, the square wound stiffness from the non-regenerating margin is ~40 kPa, while the regenerative wound center is ~13 kPa (Figure 2). Accordingly, FAK inhibitor reduced the stiffness of the entire wound bed, resulting in enhanced regeneration and WIHN (Figure 3). Conversely, actin cytoskeleton stabilizer treatment and circle wounds exhibited relatively stiffer wounds, resulting in WIHN reduction (Figure 5). This data is in agreement with Wong et al., and suggests relatively high skin stiffness during telogen and during full-thickness wound healing inhibits tissue regeneration and hair neogenesis.

An intriguing aspect is that WIHN favors a physical environment resembling anagen stiffness (Figure 2 and Appendix A). In order for hair follicular neogenesis to occur, activated epidermal cells proliferate, condense, form hair placodes, and invaginate into the dermis (30–32). Greco et al. showed that cell proliferation is an important step during wound closure, and hair follicle growth requires cell division and stem cell motility [26,36,37]. For activating epidermal cells to invaginate, these cells overcome wound stiffness (Harn 21). Thus, downregulation of ECM gene expression upon FAK inhibitor treatment and the sequential downregulation of integrin signaling fosters a relatively softer environment to enhance cell invagination for WIHN (Figure 4; Figure 5). Consequently, the FAK inhibitor treatment downregulation of ECM, integrin, and cell death gene expression and upregulation of cell motility, survival, and ephrin gene expression fosters a softer environment to enhance cell invagination and promotes WIHN. Regeneration is an intricate interaction of biochemical signals and mechanical stimulations. These cues work in concert in a spatiotemporally dynamic manner to form a favorable environment for the desired result of structure and function. This study shows how the topological distribution of mechanical cues and molecular signaling pathways work synergistically to modulate tissue response during regeneration. As a result, the production and distribution of morphogens, for example Wnt or Shh, could be affected in the wound. Whether mechanical force plays a role in regulating the distribution of these biochemical molecules and morphogens begets further study. Taken together, this study proposes a mechanism that partly explains why WIHN occurs at the wound center, and that modulation of mechanobiological cues can enhance tissue regeneration (Figure 2, Figure 4, Figure 5 and Appendix A).

Finally, the RNAseq data exhibited an increase in *Epha3* in the wound center during WIHN (Figure 4). Additional follow-up experiments showed an increase in EPHA3 in the wound center and an association of EPHA3 with pSTAT3 (Appendix A). Of note, pSTAT3 has been shown to bind the -catenin promoter to promote WIHN [4]. This manuscript confirmed pSTAT3 presence in the wound center epithelium where hair regeneration occurs during WIHN (Figure 3j). Furthermore, the data show that a reduction in wound tissue stiffness (FAK inhibitor PF573228 treatment of the entire wound) increases pSTAT3 in the wound center (Figure 3b–d,j). Taken together, these data suggest STAT3 is phosphorylated in the wound center by an additional kinase. The RNAseq data identified EphA3 as a potential candidate kinase for STAT3 in the wound center epithelium (Figure 4e). It is known that ephrin receptors are protein–tyrosine kinases and phosphorylate STAT3 [33]. The immunoprecipitation blot data showed that EPHA3 binds to pSTAT3 in the wound center epithelium (Appendix A). Taken together, this data suggests that EPHA3 phosphorylates STAT3 in the wound center epithelium during WIHN, and this is downstream and modulated by wound stiffness. Furthermore, EphA3 levels increase in the wound center epithelium during WIHN after FAK inhibitor treatment (Figure 4e). This is one reason why pSTAT3 is increased only in the wound center when FAK is inhibited. Ephrin receptors and ligands are located on adjacent cell surfaces. Ephrin receptors contain extracellular, transmembrane, and intracellular domains. including a tyrosine kinase domain that phosphorylates STAT3 (Bong 2007). A pertinent question is how the mechanical stiffness gradient signal is transmitted cell to cell during full-thickness wound regeneration and WIHN. Ephrin signaling is an interesting candidate and will be the focus of future studies.

## 5. Conclusions

In conclusion, regeneration is an intricate interaction of biochemical signals and mechanical stimulations. These cues work in concert in a spatiotemporally dynamic manner to form a favorable environment for the desired result of structure and function. This study shows how topological distribution of mechanical cues and molecular signaling pathways work synergistically to modulate tissue response during regeneration. Finally, multiple targets in the FAK/SMA/MyoII axis can be focused on for the development of regenerative medicine therapies.

## Figures and Tables

**Figure 1 pharmaceutics-14-01926-f001:**
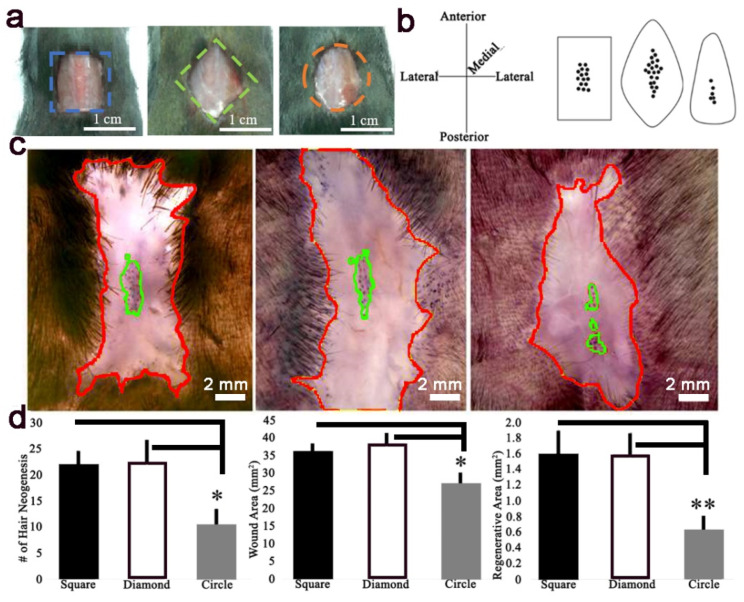
Wound-induced hair follicle neogenesis occurred in the center of all wound shapes. (**a**) Full-thickness 1 cm^2^ square, diamond, and circle wounds were created on the dorsum of p21 mice. (**b**) Summary of healed wound shapes at PWD 28 (length, anterior-posterior axis, width medial-lateral axis). (**c**) ALP staining identified regenerative dermal papillae at wound centers of square (left), diamond (center), and circle (right) (red line; wound edge, green line; WHIN area). (**d**) Quantification of WIHN, total area, and regenerative area at PWD 28 (*n* = 6. *; *p* < 0.05, **; *p* < 0.01).

**Figure 2 pharmaceutics-14-01926-f002:**
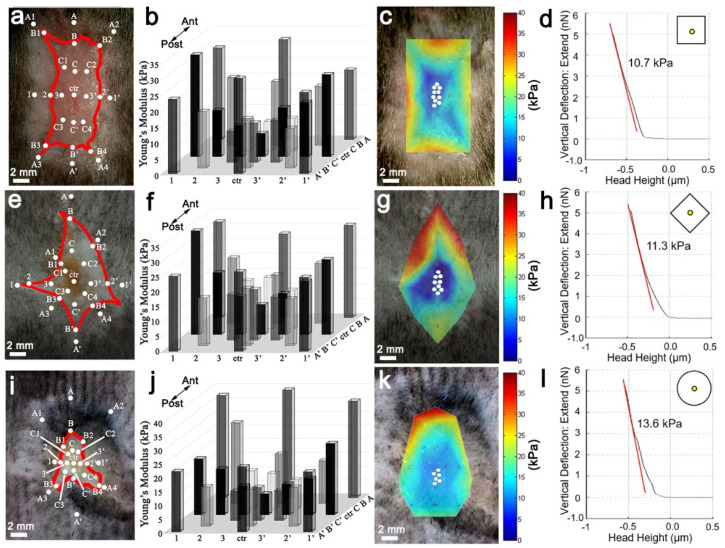
Wound-induced hair follicle neogenesis co-localized with lower stiffness. (**a**–**l**) AFM measurements for spatial distribution of stiffness at PWD 14. (**a**,**e**,**i**) Position of AFM measurements in the wound bed (white dots; AFM position, red line; wound edge, scale bar; 2 mm). (**b**,**f**,**j**) AFM quantification of square (**b**), diamond (**f**), and circle (**j**). (**c**,**g**,**k**) Overlay of heat map on PWD 14 showing average wound stiffness in square (**c**), diamond (**g**), and circle wounds (**k**) (white dots; de novo follicle positions, scale bar 2 mm). (**d**,**h**,**l**) AFM force curve measurements from PWD 14 wound centers in square (**d**), diamond (**h**), and circle (**l**) (red line; slope of linear region of the curve used to calculate Young’s modulus (kPa). Yellow dot in wound shape at top right corner of graph; force curve measurement site. The AFM cantilever spring constant is 0.08 N/m, with a 25 μm diameter bead tip, and 5 nN indentation force).

**Figure 3 pharmaceutics-14-01926-f003:**
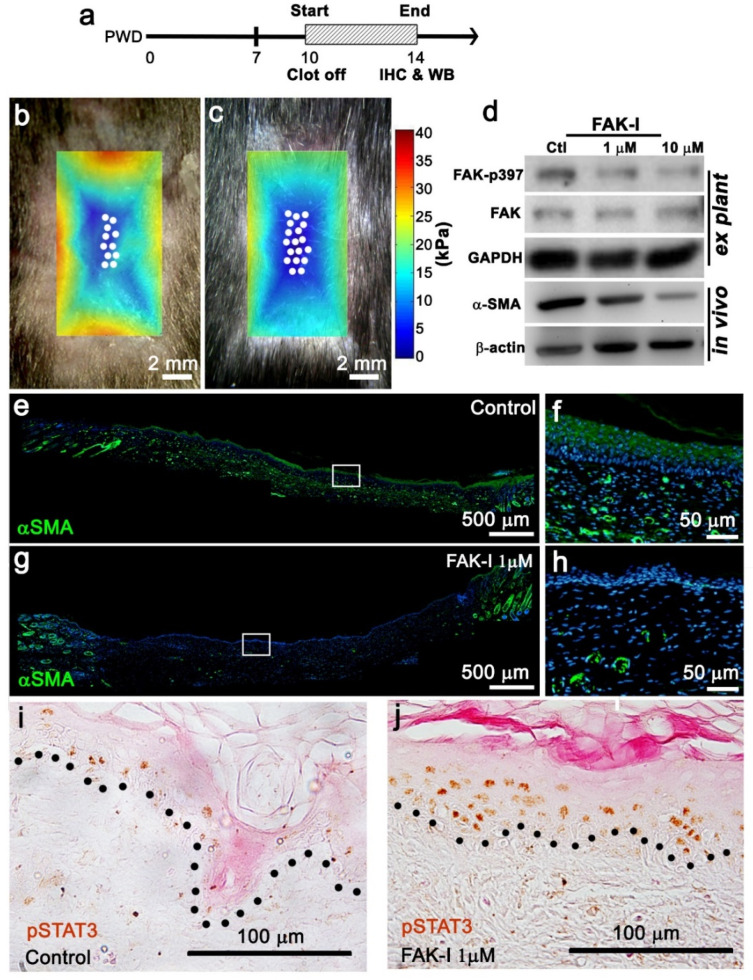
Reduced FAK activity alters the wound microenvironment resulting in enhanced wound-induced hair follicle neogenesis induction. (**a**) Timeline of topical 1 µM FAK-I treatment for western blot or immunohistochemistry experiments. (**b**,**c**) Overly of AFM heatmap onto FAK-I treated wounds exhibiting lower stiffness, increased regenerative area, and greater WIHN (scale bar; 2 mm, white dots; de novo follicle positions). (**d**) Western blot of explant wound epidermis culture or in vivo wounds treated with FAK inhibitor showing FAK, FAKp397, and α-SMA expression. (**e**–**h**) Immunostaining of topically treated 1 µM FAK inhibitor wounds showing α-SMA expression patterns (e.g., scale bar; 500 mm, f, h scale bar; 50 m, green; α-SMA, blue; Hoechst). (**i**,**j**) Immunostaining of topically treated wounds showed p-Tyr705-STAT3 expression patterns (scale bar; 100 mm, brown; p-Tyr705-STAT3, black dotted line; basal lamina).

**Figure 4 pharmaceutics-14-01926-f004:**
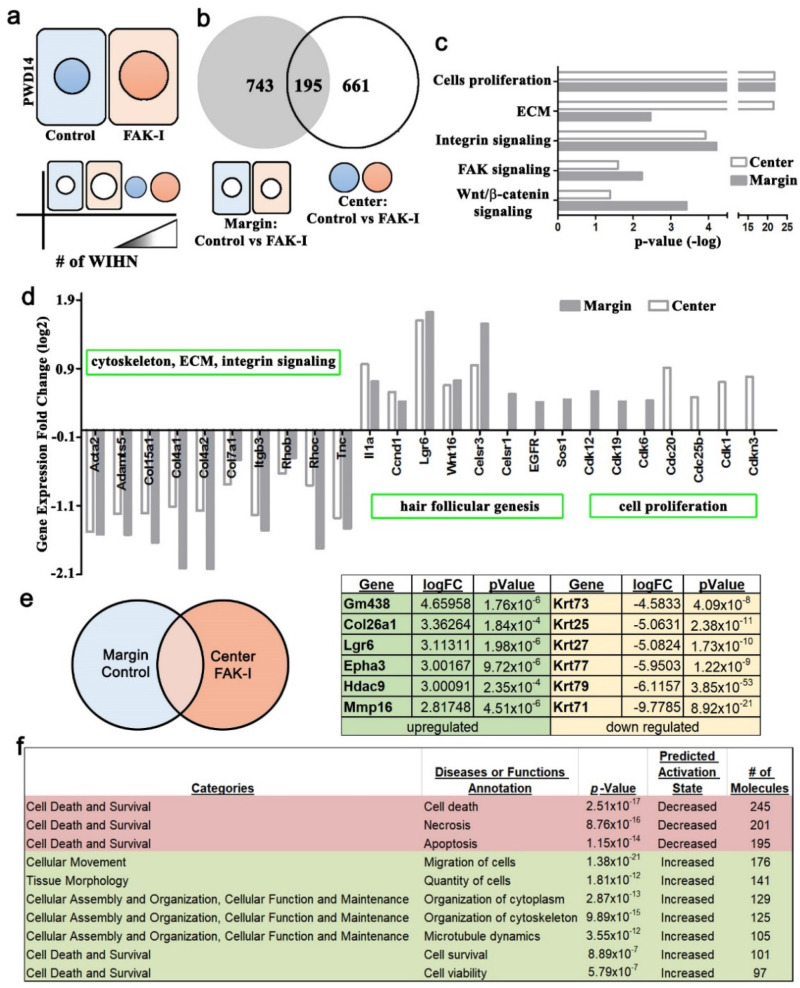
RNA-seq and GSEA identified genes and molecular pathways altered by FAK inhibition during wound-induced hair follicle neogenesis. (**a**) RNA-seq experiment diagram depicting PWD 14 wound epithelium centers and margins treated with or without 1 μM FAK inhibitor. (**b**) Venn diagram of differentially expressed genes; control versus treated center epithelium (*n* = 856) and control versus treated margin epithelium (*n* = 938). (**c**) Gene ontology analysis of control versus treated wound center or margin epithelia. (**d**) Differential gene expression analysis of control versus treated wound center or margin epithelia. (**e**) Venn diagram and table showing differential gene expression analysis of FAK-I treated center versus control margin wound epithelia highlighting top gene expression changes. (**f**) Gene set enrichment analysis of FAK-I treated wound center versus control margin wound epithelium showing upregulation of tissue morphology and downregulation of cell death functions.

**Figure 5 pharmaceutics-14-01926-f005:**
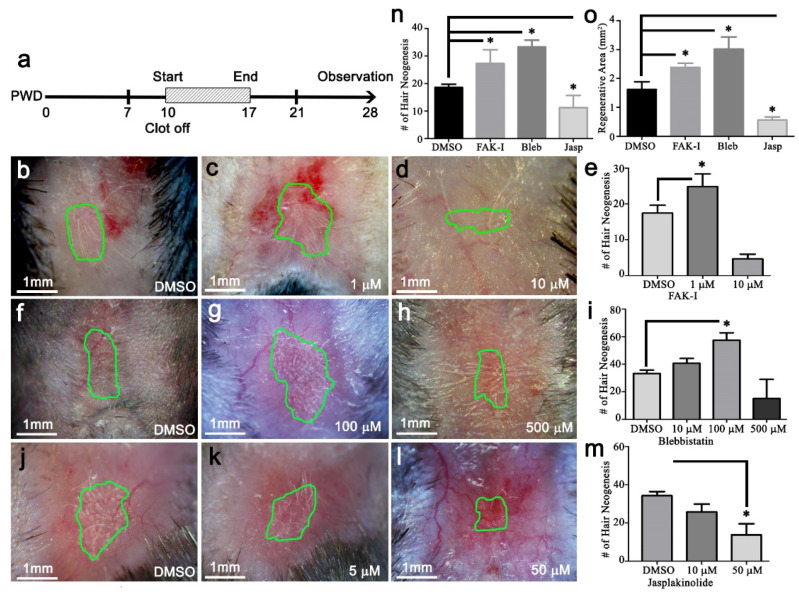
Perturbing the wound bed mechanical microenvironment modulated wound-induced hair follicle neogenesis. (**a**) Timeline of dose-response treatment experiments. (**b**–**e**) Topical treatment with FAK inhibitor PF573288 or (**f**–**i**) myosin II inhibitor blebbistatin increased WIHN and the total regenerative area at PWD 28. (**j**–**m**) Topical treatment with actin cytoskeleton stabilizer jasplakinolide treatment reduced WIHN and the total regenerative area at PWD 28. (**n**) Hair neogenesis and (**o**) regenerative area quantification after mechanotransduction perturbation treatments (green line; regeneration area outline, scale bar; 1 mm, (*n* = 4), data are represented as mean ± SEM, * *p* < 0.05).

**Table 1 pharmaceutics-14-01926-t001:** Dimensional analysis of wound geometries during wound-induced hair follicle neogenesis at PWD 28. ImageJ was used to calculate initial and final wound measurements. The circle wound exhibited the greatest reduction in healed wound area (−73.1%). There was no significant difference between the square and diamond wounds for healed wound area. The circle exhibited the shortest average final wound length (6.21 mm) and width (4.37 mm) (*i*; initial, *f*; final, Δ; change% = (final—initial)/final × 100%, data represented as mean ± SEM, (*n* = 8)).

	SQUARE	DIAMOND	CIRCLE
Area_i_	100 mm^2^	100 mm^2^	100 mm^2^
Area_f_	36.1 ± 2.2 mm^2^	38.3 ± 3 mm^2^	26.9 ± 3 mm^2^
∆ Area	−63.9%	−61.7%	−73.1%
Perimeter_i_	4.00 mm	4.00 mm	3.54 mm
Perimeter_f_	2.01 ± 0.04 mm	1.64 ± 0.07 mm	1.32 ± 0.04 mm
∆ Perimeter	−49.67%	−58.98%	−62.72%
Length_i_	10.00 mm	14.14 mm	11.28 mm
Length_f_	6.49 ± 0.32 mm	9.96 ± 0.49 mm	6.21 ± 0.72 mm
∆ Length	−35.10%	−29.60%	−44.94%
Width_i_	10.00 mm	14.14 mm	11.28 mm
Width_f_	4.17 ± 0.40 mm	4.41 ± 0.39 mm	4.37 ± 0.49 mm
∆ Width	−58.30%	−68.81%	−61.26%
Aspect Ratio_i_	1.00	1.00	1.00
Aspect Ratio_f_	1.56 ± 0.02	2.26 ± 0.60	1.42 ± 0.03
∆ Aspect Ratio	156%	226%	142%

## Data Availability

The RNAseq data has been uploaded to the Gene Expression Omnibus (GEO) website. The sample accession numbers are; GSM4850917, GSM4850918, GSM4850919, GSM4850920.

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
