# Peer review of "Topological Distribution of Wound Stiffness Modulates Wound-Induced Hair Follicle Neogenesis"

_pharmaceutics, 2022, doi:10.3390/pharmaceutics14091926_

Round 1

Reviewer 1 Report

The manuscript by Harn et al explored the possibility that the less stiffness of wound-induced hair neogenesis by the expression of p-FAK through FAK-STAT3-MyoII response, ultimately leads to the promotion of mechanotransduction in WIHN. While the data of the current study by and large support most of the claims of the manuscript, there are some concerns that need to be addressed.

1.          In material and methods, the authors claimed to use tdTomato-EGFP mouse in vivo assay, however, there is no following data in the results. Please revise or show the relevant results.

3.          The authors disclosed that Wound shape affects hair follicle neogenesis and correlates with lower stiffness which regulates by p-FAK (397). What is the p-FAK (397) expression pattern in different wound shape-induced WIHN? Does it only express in a certain cell type?

4.          In Fig3C and G, what is the dosage of FAK- inhibitor? Why do the authors use that dose? Same question on Fig 5N and 5O. Can authors provide FAK-p397 in both explant and in vivo?

5. Fig 3J, in the image, does FAK-I 1mM mean the inhibitor use 1mM? If yes, why not the dose is not consistent in the following experiments? What is the phosphate site of STAT3? As known, STAT3 is a downstream target of FAK, which is positively regulated by FAK. How come after blocking the FAK, p-STAT3 increased in Fig3J.?

6.          The authors showed that FAK results in stiffness, why not conditional knockout FAK then monitor the stiffness and the hair neogenesis in WIHN?

7.          In Fig 5, why use a high dose of FAK inhibitor and blebbistatin reduced hair neogenesis while a low dose dramatically promotes hair regeneration? Can authors measure the stiffness by AFM in those high doses treated groups?

8.          Extracellular matrix contributes to stiffness. Besides, the RNA seq data showed the extracellular matrix gene significantly changed. Have the authors checked which extracellular matrix proteins changed most?

9.          FAK as a focal adhesion complex protein, do other proteins also change? E.g. p-Src, talin, Vinculin, paxillin?

10.        Fig 3 figure legend, is that “FAKp357” should be FAKp397?

11.        The authors claimed that EPHA3 binds p-STAT3, thus CO-IP EPHA3 should add a negative control. Can authors provide more data to prove they are binding? PLA assay or immunofluorescence staining assay may help to confirm. 

Reviewer 2 Report

In the manuscript entitled "Topological distribution of wound stiffness modulates wound induced hair follicle neogenesis", the authors analyzed the wound stiffness of different wound shapes and investigate its influence on the hair follicle neogenesis. This is an very interesting topic and the presented data are convicing.

In Figure 3d, different endo references were used in explant and invivo samples. Why not use a same one?

The modulation pattens and mechanisms involved are largely unknown. RNA-seq were used in this study to provide clues. Spatial transcriptomics/single cell sequencing could be applied to get more.

Round 2

Reviewer 1 Report

The authors have addressed my concerns well from my previous review. They have responded to all my questions and made the necessary changes to the manuscript.